# Investigation of Phosphatidylserine-Transporting Activity of Human TMEM16C Isoforms

**DOI:** 10.3390/membranes12101005

**Published:** 2022-10-17

**Authors:** Hanggu Kim, Eunyoung Kim, Byoung-Cheol Lee

**Affiliations:** Neurovascular Unit Research Group, Korea Brain Research Institute (KBRI), Daegu 41068, Korea

**Keywords:** membrane proteins, lipid transport, scramblases, ion channels

## Abstract

Lipid scrambling is a rapid process that dissipates the asymmetrical distribution of phospholipids in the plasma membrane. It is involved in various physiological functions such as blood coagulation and apoptosis. Many TMEM16 members are recognized as Ca^2+^-activated phospholipid scramblases, which transport phospholipids between the two leaflets of the plasma membrane nonspecifically and bidirectionally; among these, TMEM16C is abundant in the brain, especially in neuronal cells. We investigated the scrambling activity of three human TMEM16C isoforms with different N-terminus lengths. After optimizing conditions to minimize endogenous scrambling activity, an annexin V-based imaging assay was used to detect phosphatidylserine (PS) scrambling in 293T cells. Unlike previous results, our data showed that human TMEM16C isoform 1 and isoform 3 exposed PS to the cell surface. A surface biotinylation assay showed that the surface expression of isoform 2, which did not show scrambling activity, was ~5 times lower than the other isoforms. In contrast to other TMEM16 proteins, flux assays and electrophysiology recording showed TMEM16C does not possess ion-transporting activity. We conclude that the N-terminus of TMEM16C determines whether TMEM16C can translocate to the plasma membrane and facilitate scrambling activity; membrane-localized TMEM16C isoforms 1 and 3 transport PS to the outer leaflet.

## 1. Introduction

Lipid scrambling is a process that changes the asymmetric distribution of phospholipids in the outer- and inner leaflets of cell membranes [1]. Whereas phosphatidylcholine and sphingomyelin largely reside in the outer leaflet, the inner leaflet is composed primarily of phosphatidylserine (PS) and phosphatidylethanolamine. This asymmetry is created and maintained by two ATP-driven pumps called flippase and floppase [1,2]; the enzyme that scrambles or mixes lipids in the membrane is called “scramblase”. Lipid scrambling is used as a signal by cells [2], the most well-known being the “eat me” signal from apoptotic cells. In normal cells, PS is dominant in the inner leaflet; however, lipid scrambling causes dying cells to expose PS on the outer membrane, a change recognized by phagocytic cells, which then remove apoptotic cells [3]. Lipid scrambling is involved in various other physiological processes such as blood coagulation, synaptic pruning, viral infection, fertilization, and myoblast fusion [4].

There are three types of lipid scramblases in cell membranes, which are classified by their activation mechanism. The first, Ca^2+^-activated lipid scramblases, are activated by increased intracellular Ca^2+^ [5]. The second type, the Xk-related (Xkr) family, are activated by caspase signals; they contain caspase recognition domains and their cleavage by caspase is essential for scrambling activity [6]. The final class, constitutively active or non-regulated lipid scramblases, includes G protein-coupled receptors such as Rhodopsin [7,8]. In recent decades, several studies have tried to determine the molecular identity of Ca^2+^-activated lipid scramblases, with members of the TMEM16 family now accepted as belonging to this class [9]. 

TMEM16 proteins are composed of 10 family members and were initially reported as Ca^2+^-activated Cl^−^ channels [10,11,12]. Despite the similarity in their genetic sequence, TMEM16 proteins are functionally divergent. For instance, TMEM16A and TMEM16B are recognized as Ca^2+^-activated Cl^−^ channels, but TMEM16C, TMEM16D, TMEM16E, TMEM16F, TMEM16G, TMEM16J, and TMEM16K have Ca^2+^-activated lipid-scrambling activity [13,14,15,16]. Among these, TMEM16E, TMEM16F, and TMEM16K are dual-function, non-selective ion channels and lipid scramblases [14,15,16]. Structural studies show they are dimeric proteins with 10 transmembrane domains [15,17,18,19,20]. TMEM16 proteins have a primary Ca^2+^ binding site within the transmembrane region and an additional binding site near the dimeric interface. Among TMEM16 family members, TMEM16C is mainly found in the brain [13,21,22], especially in neuronal cells [23,24]. TMEM16C does not have ion- or PS-transporting activity, but does transport phosphatidylcholine and galactosylceramide [13]. TMEM16C also regulates pain-related signals in rat dorsal root ganglia by modulating the activity of sodium-activated potassium channels [24,25]; although the mechanism is not well understood, changes in lipid distribution by the lipid-scrambling activity of TMEM16C may be involved. 

TMEM16C proteins are associated with many neuronal diseases, including febrile seizure [26] and autosomal-dominant craniocervical dystonia [22]. TMEM16C is also associated with late-onset Alzheimer’s disease [27]. Interestingly, transcriptional analysis shows that TMEM16C is downregulated in patients suffering from Alzheimer’s disease compared with healthy controls [28]. These results suggest that TMEM16C could be a new target for the treatment of brain disease. 

In this study, we investigated the lipid-scrambling and ion-transporting activity of human TMEM16C isoforms. Among three human isoforms, isoform 1 and 3 transported PS to the membrane outer leaflet, while isoform 2 did not. We confirmed that these results were due to differences in the surface expression level of each isoform. We also found that GFP tagging on isoform 1 severely reduced the scrambling activity of TMEM16C. Ion channel activity of the TMEM16C isoforms was not observed in either flux assays or in the electrophysiological recording. These results suggest that human TMEM16C isoforms that present on the surface membrane are PS-transporting lipid scramblases but not ion channels.

## 2. Materials and Methods

### 2.1. Construction and Expression of TMEM16C Isoforms

A human, cDNA ORF clone of TMEM16C isoform 1 (NCBI, NP_001300655.1) was purchased from GenScript (Clone ID, Ohu71184). Full-length DNAs of isoform 2 (NP_113606.2) and isoform 3 (NP_001300656.1) were generated by PCR and cloned into a pCEP4 vector (Thermo). For GFP tagging of each DNA construct, pCAG-GFP (Addgene, #11150) vectors were used. To estimate transcription efficiency, pCAG-GFP virgin vectors were co-transfected with TMEM16C constructs at a molar ratio of 1:100 (GFP:TMEM16C). To confirm the presence of TMEM16C isoforms in the human brain, whole brain QUICK-Clone cDNA (TaKaRa) and PCR were used with isoform 1 and isoform 2-specific primers, namely isoform 1 forward primer: ATGTCAGTTTTAAAATTTGAACTG; isoform 2 forward primer: ATGGTCCACCATTCAGGCTCCATT; isoform 3 forward primer: GCAATGAAGGATTCCAAATGCAGCTG; and universal reverse primer: TCCCCCCGGGGAGGCCATTCATGGTG.

### 2.2. Scrambling Assay

24 to 48 h after transfection using X-tremeGENE HP (Roche), scrambling activity was measured in 293T cells. A 0.5% Alexa Fluor 568-tagged annexin V (Invitrogen) solution was prepared by making a scrambling solution (140 mM NaCl, 10 mM CaCl_2_, 10 mM HEPES, pH 7.4). To activate phospholipid scramblases, an equal volume of scrambling solution containing 4 μM of the Ca^2+^ ionophore A23187 (Sigma) was added to reach a final concentration of 2 μM. As an unstimulated control, 0.1% DMSO was added to cells in annexin V-binding solution. Exposure of PS on the outer cell membrane was measured by the accumulation of annexin V–Alexa Fluor 568 and captured with time-lapse imaging and a super-sensitive, high-resolution confocal laser scanning microscope TCS SP (Leica) for 20 min after the addition of the ionophore. Image acquisition was controlled by LAS X software (Leica). 

### 2.3. Surface Biotinylation and Western Blot

For the surface biotinylation of TMEM16C isoforms, 293T cells were seeded at a density of 6 × 10^5^ cells in a 6-well plate. Cells were transfected with the DNA of TMEM16C isoforms and incubated for 48 h. During the experiment, the plate was fixed on ice. 293T cells were washed three times with PBS and incubated with 2 mL of biotinylation solution containing EZ-LinkTM-sulfo-NHS-SS-Biotin (Thermo) (0.35 mg/mL in PBS) for 20 min. Next, an ice-cold PBS buffer containing 50 mM glycine was added to stop the reaction. After an additional incubation of 5 min, cells were collected by centrifugation at 4000× *g* for 1 min. Cells were washed with PBS and lysed with lysis buffer containing 1% Triton X-100 and protease inhibitor cocktail by inverting for 1 h at 4 °C. The supernatant was collected and biotinylated proteins were bound to Streptavidin Plus UltraLinkTM Resin (Thermo) for 3 h at 4 °C. After this incubation period, samples were spun down and washed three times with 1 mL of lysis buffer. The pellet of streptavidin beads was resuspended in 40 μL of 3× LDS buffer for 10 min. Protein samples were then loaded onto the protein gel and a western blot was conducted with target antibodies, TMEM16C (Human protein atlas), actin (Cell signaling) and transferrin receptor (abcam). Relative surface expression ratios were calculated by normalizing the band intensity of TMEM16C in the surface to the band intensity of TMEM16C in the total (surface/total).

### 2.4. Halide-Quenching Flux Assay

To measure the ion-transporting activity of TMEM16C isoforms using a cell-based assay, YFP-H148Q/I152L-stable cell lines were used. YFP-H148Q/I152L in pcDNA3.1 was transfected into 293 cells and stable cells were selected using G418 antibiotics. YFP-expressing 293 cells were seeded into 96-well plates and TMEM16C isoforms and human TMEM16A were transfected with X-tremeGENE HP (Roche). Human TMEM16A (NCBI, XM_011545127.3) was used as a positive control. Prior to measurement of the fluorescence signal, the medium was changed to 50 μL basal buffer (137 mM NaCl, 2.7 mM KCl, 2 mM CaCl_2_, 1 mM MgCl_2_, 10 mM HEPES, pH 7.4). The signal was acquired using the Flexstation 3 (Molecular device) at an excitation and emission wavelength of 513 nm and 527 nm, respectively. After stabilization of the signal for 60 s, 100 μL of iodide-containing buffer (137 mM NaI, 2.7 mM KCl, 2 mM CaCl_2_, 1 mM MgCl_2_, 10 mM HEPES, pH 7.4) was added to each well to monitor iodide-transporting activity. Iodide buffers both with and without the Ca^2+^ ionophore (10 μM) were prepared.

### 2.5. Electrophysiology

The macroscopic current of TMEM16C isoforms was recorded by whole-cell configuration. Patch pipettes were fabricated from borosilicate glass and each resistance was between 2–4 Mohm. The currents were amplified using an Axopatch 200B (Molecular Device) and filtered at 2 kHz with a lowpass Bessel filter. Signals were digitized at a rate of 5 kHz using an Axon Digidata 1550B digitizer (Molecular Device). Ionic currents were evoked by voltage stimulus delivered from a holding potential of 0 mV to test voltages ranging from −100 to 100 mV. For the activation of TMEM16C isoforms, recording conditions described previously were used [29]. The intracellular solution contained 146 mM CsCl, 2 mM MgCl_2_, 5 mM EGTA, 10 mM Sucrose, 10 mM HEPES, pH 7.3. Based on the MaxChealator (https://somapp.ucdmc.ucdavis.edu/ (accessed on 3 May 2022), free Ca^2+^ concentration was calculated and the required amount of CaCl_2_ was added. The extracellular solution contained 140 mM NaCl, 5 mM KCl, 2 mM CaCl_2_, 1 mM MgCl_2_, 15 mM Glucose, 10 mM HEPES, pH 7.4. The pH of both solutions was titrated with NMDG. As a positive control, TMEM16F (NCBI, NM_001025356.3) isoform 1 was used to measure the macroscopic currents activated by intracellular Ca^2+^. For each data set, the significance of the difference was tested using a paired sample t-test using Origin software (Originlab). In all cases, *p* < 0.05 was considered significant.

## 3. Results

### 3.1. Human TMEM16C Isoforms Contain Different N-Terminuses

Most studies on TMEM16C are conducted using mouse [13,24] and rat TMEM16C [30]. Mouse and rat isoform 1 are almost identical (Figure 1A,B) and correspond to human TMEM16C isoform 2. In humans, there are two further isoforms of TMEM16C: isoform 1 is a novel TMEM16C isoform that contains a longer N-terminal segment than mouse and rat TMEM16C (Figure 1A); isoform 3 contains the shortest N-terminus and corresponds to mouse isoform 2 (Figure 1A,B). To identify whether isoform 1 is present in the human brain at the transcriptional level, we obtained commercially available cDNA from human brain. PCR demonstrated that isoform 1 of human TMEM16C could be amplified using human isoform 1-specific primers and the amount of amplified DNA was less than that of isoform 2 (Figure 1C, left). Since the CDS (Coding Sequence) of isoform 3 is perfectly matched to that of isoform 1, isoform 3-specific primer was designed in the UTR (Untranslated Region) of isoform 3. Finally, we could observe isoform 3-specific PCR fragments in the gel (Figure 1C, right). These results suggest that all three human TMEM16C isoforms exist in the human brain at the transcriptional level.

### 3.2. Endogenous Scrambling Activity of the 293T Cell and the Effects of GFP Tagging on the Scrambling Activity

To measure scrambling activity, we conducted an optical imaging, cell-based assay to monitor exposure of PS on cell membranes. In unstimulated conditions, most PS exists in the inner leaflet of the plasma membrane (Figure 2A, left). Treatment with the Ca^2+^ ionophore A23187 increases intracellular Ca^2+^, allowing Ca^2+^-activated lipid scramblases to be activated for transporting PS to outer leatlet. Since annexin V could bind to PS specifically, PS exposed on the outer leaflet can then be visualized by fluorophore-conjugated annexin V (Figure 2A, right). Thus, accumulation of the fluorescent signal can be monitored using real-time imaging.

Prior to measuring the scrambling activity of TMEM16C isoforms, the endogenous scrambling activity of 293T cells was investigated. As reported previously [31], 293T cells showed scrambling activity after 10 μM ionophore treatment (Figure 2B), with 23.3% of cells showing scrambling activity (Figure 2E). The control treatment, 0.1% DMSO, did not affect the scrambling process in 293T cells (Figure 2B). To find the optimal conditions to prevent endogenous TMEM16F activity from interfering with the measurement of TMEM16C activity, we tested the effect of various concentrations of a Ca^2+^ ionophore on the scrambling activity of 293T cells. We concluded that no scrambling activity was observed in the presence of 10 mM CaCl_2_ and 2 μM ionophore (Figure 2C,E). 

To confirm transfection with our genes of interest, we used a GFP-tagged construct, which has been used previous in studies measuring the scrambling activity of TMEM16F [16,32]. However, in the presence of 2 μM ionophore, scrambling activity was only observed in 2.2% and 0.9% of cells expressing isoform 1-GFP and isoform 2-GFP, respectively (Figure 2D,F). These results suggest that a very small number of cells expressing GFP-tagged TMEM16C proteins responded to an increment of intracellular Ca^2+^, consistent with a previous result that showed that tagged mouse TMEM16C displayed minimal PS scrambling activity [13]. 

### 3.3. Scrambling Activity of Three TMEM16C Isoforms

Since the ion-and lipid-transporting activity of nhTMEM16, a fungal homologue of TMEM16, is inhibited by GFP tagging [33], we next probed the effect of GFP tagging of the scrambling activity of TMEM16C isoforms. We generated constructs without a GFP tag and repeated the cell-based imaging assay. To validate the transfection of the genes and expression of the TMEM16C, DNA expressing GFP was co-transfected with TMEM16C isoforms at a molar ratio of 1:100; we assumed that cells expressing GFP were also expressing TMEM16C isoforms. In each case, transfection efficiency was similar in ionophore- (red bars) and DMSO-treated (control; black bars) cells (Figure 3A–C): 41.1% and 43.7% for isoform 1, 33.8% and 31.9% for isoform 2, and 34.9% and 40.1% for isoform 3, respectively (Figure 3D).

Unlike with the GFP-tagged protein, 23.5% of cells expressing isoform 1 without a GFP tag presented PS on the outer leaflet in the presence of 2 μM ionophore (Figure 3A,E). These results suggest that GFP tagging was inhibiting the scrambling activity of isoform 1, as shown in a previous study which was performed with fungal TMEM16 homologues, nhTMEM16 [33]. Consistent with a previous study [13], isoform 2, which corresponds to mouse isoform 1, showed very low activity, with 2.8% of cells displaying PS on the outer leaflet (Figure 3B). Isoform 3, the shortest TMEM16C isoform, also transported PS to the outer membrane (Figure 3C). Upon the increment of intracellular Ca^2+^, 10.3% of 293T cells expressing TMEM16C isoform 3 transported PS (Figure 3E). All cells were also exposed to a control condition (0.1% DMSO) without Ca^2+^ addition. For isoform 2 and 3 of TMEM16C, 0.1% DMSO did not stimulate any scrambling activity of 293T cells; however, in cells expressing isoform 1, 0.1% DMSO resulted in the scrambling activity in 0.5% of cells (black bars, Figure 3E). This may result from the expression of isoform 1 affecting cell viability, causing cells to generate the apoptotic “eat me” signal which would also be stained with the annexin V-based assay.

### 3.4. Surface Expression of Human TMEM16C Isoforms

These results suggested that the isoforms with the longest (isoform 1) and shortest (isoform 3) N-terminuses can transport PS, while the isoform of intermediate length N-terminus (isoform 2) cannot. Thus, we next investigated the surface expression level of each isoform in 293T cells by conducting a surface biotinylation assay. No differences in the expression of each isoform was observed when the expression levels of the isoforms were compared using the immunoblotting of total cell lysates, (Figure 4A). However, a surface biotinylation assay showed much lower expression of isoform 2 on the cell surface than isoform 1 or 3: the relative surface expression ratios of isoform 1 and 3 were 0.46 and 0.45, respectively, while that of isoform 2 was 0.10 (Figure 4B,C). The actin and transferrin receptors were used as a control protein for cytosolic- and membrane proteins, respectively. Immunoblotting results for the transferrin receptor and actin showed that only proteins on the cell surface are biotinylated (Figure 4B). These results suggest that the N-terminal region of TMEM16C is critical for the translocation of TMEM16C to the plasma membrane. Additionally, we tested the surface expression of the GFP tagged TMEM16C isoforms by conducting surface biotinyaltion assays with a GFP tagged construct. Unlike untagged isoforms, GFP tagged isoform 3 showed somewhat lower expression in total fraction (Figure 4C). The relative surface expression ratios of GFP tagged isoform 1, 2, and 3 were 1.63, 0.52, and 0.52, respectively. These results suggest that GFP tagged TMEM16C isoform 1 and 2 are more abundant in the cell surface than untagged constructs. 

### 3.5. Ion Channel Activity of Human TMEM16C Scramblases

Next, we studied the ion-transporting activity of TMEM16C isoforms. Previous studies suggest that the heterologous expression of mouse TMEM16C isoform 1, which corresponds to human isoform 2, does not result in ion-transporting activity [13,24]. Prior to conducting electrophysiology studies, we measured ion transportation using a YFP-based halide ion-quenching assay. Since fluorescent signals from YFP(H148Q/I152L) could be quenched by the halide ion, this assay was largely used in the study on the anion transporting activity of ion channels. After establishing YFP (H148Q/I152L)-expressing stable cell lines, cells were transfected with DNA for each TMEM16C isoform. In all cases, the iodide treatment of TMEM16C-expressing cells in the presence of 10 μM ionophore did not decrease the fluorescent signal compared with untransfected cells (Figure 5A). However, cells expressing human TMEM16A, a well-known Ca^2+^-activated Cl^−^ channel, showed the decrease in the YFP signal after treatment of Ca^2+^ ionophore and iodide (Figure 5A). To measure the electrical activity of human TMEM16C directly, electrophysiological recordings were performed using whole-cell configuration. At 2 μM intracellular Ca^2+^, no TMEM16C isoforms showed any changes in the ionic current compared to untransfected cells (Figure 5B). Since the purpose of patch clamp recording was to determine whether TMEM16C isoforms have a channel function or not, a high concentration of free Ca^2+^ (200 μM) was also tested in the intracellular solution, as reported in other studies [29,31]. Compared with untransfected cells, TMEM16C isoform-expressing cells did not show any significant changes in the whole-cell current upon Ca^2+^ and voltage stimulus (Figure 5C). As a positive control for validating our assay system, human TMEM16F isoform 1 was also transfected and their electrical currents were measured in the presence of 200 μM Ca^2+^. The cells expressing human TMEM16F showed large macroscopic currents (Figure 5D). All current traces recorded from human TMEM16C isoforms were almost similar to the endogenous currents from untransfected 293T cells. We could only observe the endogenous outward membrane currents. These currents were activated with voltage-dependent activation kinetics upon higher positive voltage stimulus (Figure 5B,C). These activation kinetics over time were also shown in the current traces of TMEM16F transfected cells. The current density from all isoforms was comparable with that of untransfected 293T cells (Figure 5D), and these values are statistically insignificant. These results suggest that TMEM16C does not have ion-conducting activity. 

## 4. Discussion

To date, many studies measuring ion-transporting and/or the lipid-scrambling activity of the TMEM16C protein have used tagged mouse or rat TMEM16C isoform 1. For instance, the Nagata group used mouse isoform 1 with C-terminal FLAG tagging [13], and the Jan group measured the activity of mouse isoform 1 with N-terminal HA tagging and C-terminal GFP tagging [24]. Several studies have used GFP or YFP-tagged constructs to investigate other members of the TMEM16 family, such as TMEM16A and TMEM16F [12,16,17,32], without observing an effect of the fluorescent tagging on the protein’s function. However, we previously examined the effect of GFP tagging on the activity of nhTMEM16, a fungal homologue of TMEM16 protein, and found that GFP tagging inhibited both its ion- and lipid-transporting activity [33]. Likewise, the present study suggests that GFP tagging of human TMEM16C isoform 1 severely reduces PS-transporting activity. After removing the GFP tag from the isoform 1 construct, the percentage of scrambled cells increased from 2.2% to 23.5%; for isoform 2, a slight increase from 0.9% to 2.3% scrambling was observed after GFP tag removal. To determine whether the reduction of scrambling activity of GFP tagged TMEM16C is caused by the inhibition of protein function itself or due to the defects on the surface expression of the protein, we conducted a surface biotinylation assay by using GFP tagged constructs. We found that more TMEM16C isoform 1 and 2 exist in the cell surface after adding the GFP tag. These results strongly suggest that GFP tagging on isoform 1 could inhibit the PS transporting activity without inhibiting the translocation of isoform 1 to the cell surface. For isoform 2, even though the scrambling activity was slightly increased after cleavage of the GFP tag, GFP tagging inhibited the PS transporting function not by inhibiting its translocation to the cell surface.

Previous findings from studies of mouse TMEM16C show that this protein did not transport PS in the presence of the Ca^2+^ increment [13]. Consistent with this result, our experiments showed that human isoform 2, corresponding to mouse isoform 1, did not transport PS to the cell surface because its surface expression was significantly lower than that of other isoforms; however, human isoforms 1 and 3 transported PS, similar to TMEM16F. The exposure PS to the outer leaflet is a representative consequence of lipid scrambling. Since the most basic feature of lipid scramblases is non-selective lipid transport, the result that showed that TMEM16C discriminated between PS and other lipids, phosphatidylcholine and ceramide was unexpected [13]. For this reason, interest in TMEM16C as a lipid scramblase has declined despite its involvement in neurological diseases. Thus, our finding provides evidence that certain TMEM16C isoforms possess the predominant feature of lipid scramblases, PS-transporting activity. Furthermore, we measured the PS-transporting activity of TMEM16C in the presence of a lower Ca^2+^ concentration than that required to activate endogenous human TMEM16F in 293T cells. These results suggest that TMEM16C could have a role in cell surface PS exposure in response to small increases in intracellular Ca^2+^. 

In summary, we investigated the lipid- and ion-transporting activity of three human TMEM16C isoforms using optical imaging, a fluorophore-quenching flux assay, and electrophysiological recording. We showed that isoform 1 and isoform 3 could transport PS to the outer leaflet, while isoform 2 could not, due to significantly lower isoform 2 surface expression. We could not detect macroscopic ionic currents from surface-expressed TMEM16C isoforms 1 and 3. Taken together, these results suggest that human TMEM16C present on cell surface membranes is a PS-transporting lipid scramblase but not an ion channel.

## Figures and Tables

**Figure 1 membranes-12-01005-f001:**
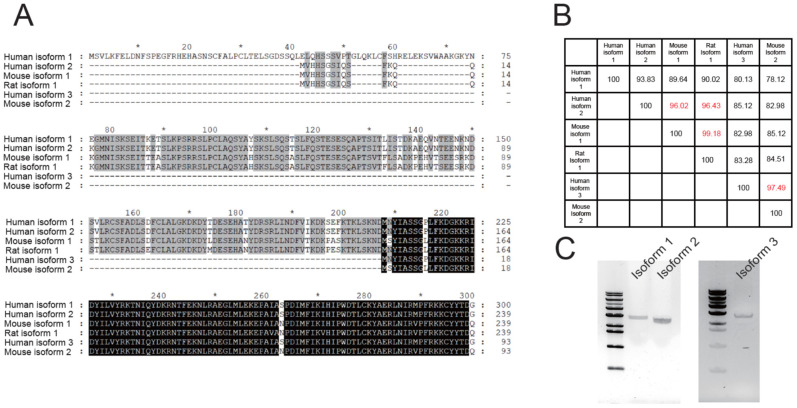
Three human TMEM16C isoforms and their similarity to other orthologues. (**A**) Sequence alignment of the N-terminal regions of human, mouse and rat TMEM16C isoforms. Alignment was conducted using Clustal Omega. (**B**) Sequence similarity between human TMEM16C isoforms and orthologues; similarity was calculated using the Ident and Sim program from the Sequence Manipulation Suite. (**C**) PCR results for human TMEM16C isoforms using cDNA from human brain. The first column in the gels is 1 kb DNA ladder to estimate the length of DNA.

**Figure 2 membranes-12-01005-f002:**
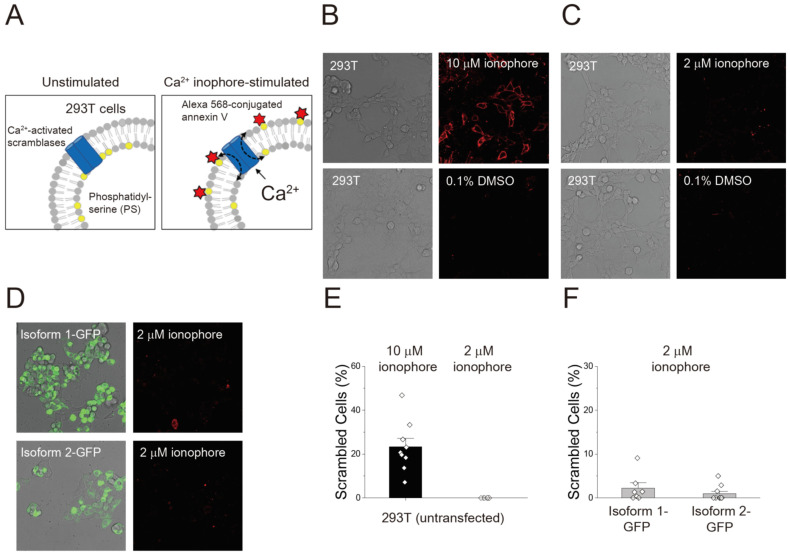
Scrambling activity of endogenous TMEM16F in 293T cells and the effect of GFP tagging on human TMEM16C. (**A**) Schematic diagram of cell-based imaging using fluorophore-labeled annexin V. Phosphatidylserine (PS) exposed by Ca^2+^-activated scramblase activity upon treatment with the Ca^2+^ ionophore A23187 was stained with Alexa 568-conjugated annexin V. Accumulation of red signal was captured by microscope imaging. (**B**) Endogenous scrambling activity of TMEM16F in 293T cells. 10 μM ionophore was added to 293T cells to increase intracellular Ca^2+^ concentration. DMSO (0.1%) was used as a control. (**C**) Optimization of imaging conditions to minimize endogenous lipid-scrambling activity in 293T cells. 293T cells treated with 2 μM ionophore showed minimal scrambling activity, equivalent to the control treatment. (**D**) Lipid-transporting activity of GFP-tagged human TMEM16C isoforms 1 and 2. 2 μM ionophore was added to transfected 293T cells to increase intracellular Ca^2+^ and stimulate TMEM16C activity. (**E**) Quantification of lipid-scrambling activity in 293T cells upon treatment with 10 μM and 2 μM ionophore. Data are presented as mean + standard error of the mean (SEM); n = 9 for 10 μM ionophore and n = 8 for 2 μM ionophore. (**F**) Quantification of lipid-scrambling activity of GFP-tagged TMEM16C isoform-transfected 293T cells after treatment with 2 μM ionophore. Data are presented as mean + SEM; n = 7 for isoform 1-GFP and n = 10 for isoform 2-GFP.

**Figure 3 membranes-12-01005-f003:**
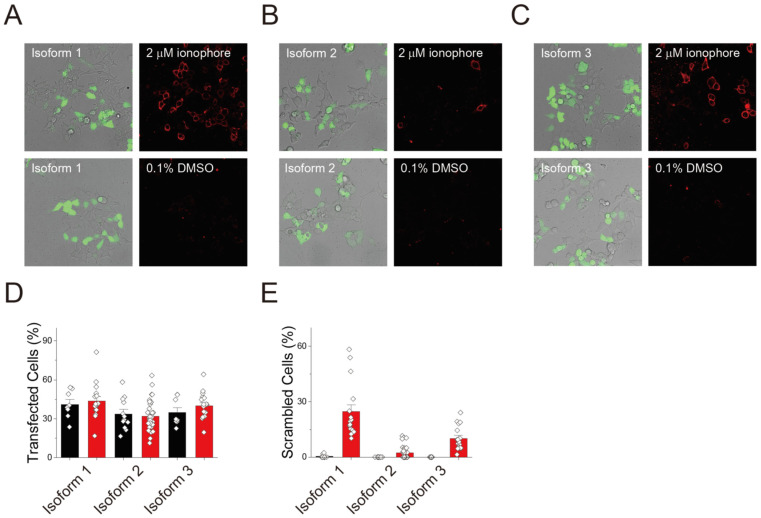
Scrambling activity of three human TMEM16C isoforms without GFP tagging. Monitoring of the lipid-transporting activity of TMEM16C isoform 1 (**A**), isoform 2 (**B**), and isoform 3 (**C**). A total of 293T cells were transfected with each isoform, without a GFP tag. To confirm transfection, a pCAG-GFP construct was co-transfected with TMEM16C at a ratio of 100:1 (TMEM16C:GFP). 2 μM ionophore was used to increase intracellular Ca^2+^ concentration; DMSO (0.1%) was used as a control. (**D**) Transfection efficiency of each isoform. The percentage of transfected cells was calculated for ionophore- (red bars) and DMSO-treated cells (black bars) by counting the total cells in brightfield images and GFP-expressing cells in fluorescence images. (**E**) Quantification of scrambling activity of human TMEM16C isoforms after treatment with a control (DMSO) or 2 μM ionophore. Data are presented as mean + SEM. Isoform 1: DMSO, n = 9 and ionophore, n = 16; isoform 2: DMSO, n = 12 and ionophore, n = 34; isoform 3: DMSO, n = 8 and ionophore, n = 17.

**Figure 4 membranes-12-01005-f004:**
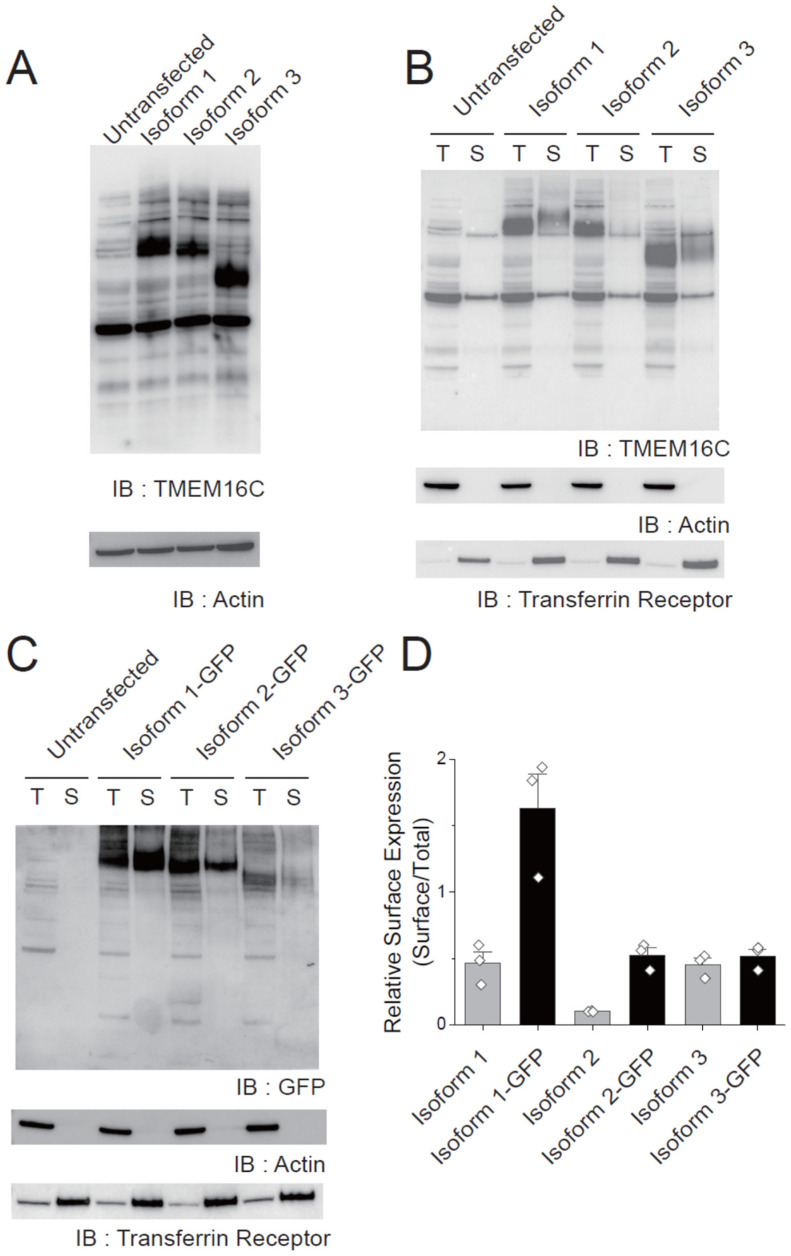
Surface expression of human TMEM16C isoforms. (**A**) Immunoblotting of cell lysates expressing human TMEM16C isoforms. Anti-TMEM16C (upper image) and anti-actin (lower image) antibodies were used. (**B**) Immunoblotting of surface biotinylated TMEM16C proteins. Surface fraction (S) samples were loaded at a 10-fold higher concentration than the total fraction (T). Anti-TMEM16C (upper image), anti-actin (middle image) and anti-transferrin receptor (lower image) antibodies were used to validate the biotinylation of surface-expressed proteins. (**C**) Immunoblotting of surface biotinylated GFP tagged TMEM16C proteins. (**D**) Quantification of relative surface expression of each isoform. Relative surface expression ratios were calculated by normalizing the band intensity of TMEM16C in the surface to the band intensity of TMEM16C in the total. Mean + SEM; n = 3 for each isoform.

**Figure 5 membranes-12-01005-f005:**
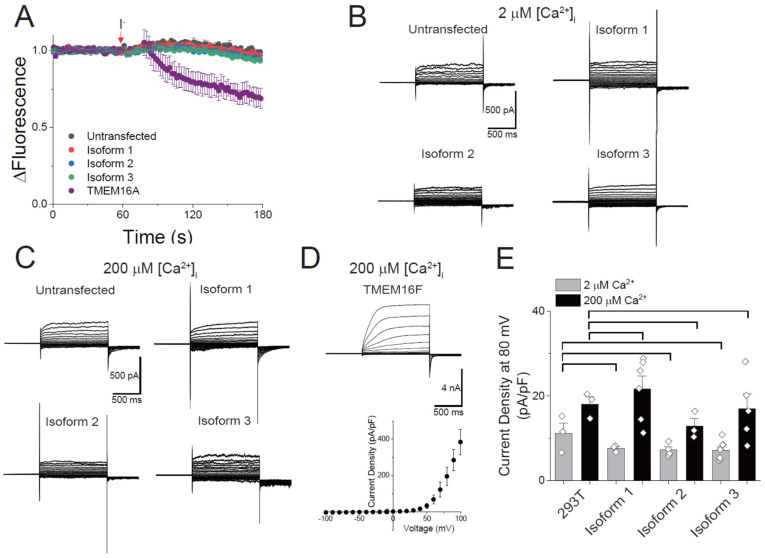
Ion-transporting activity of human TMEM16C isoforms. (**A**) Measurement of ion-transporting activity using an iodide-quenching assay. Each TMEM16C isoform and human TMEM16A were transfected into 293T cells stably expressing YFP-H148Q/I152L. Mean + SEM; n = 3. (**B**,**C**) Representative whole-cell current recordings of cells transiently transfected with human TMEM16C isoforms. 2 μM (**B**) and 200 μM (**C**) intracellular Ca^2+^ was added and ionic currents were evoked with voltage steps ranging from −100 mV to +100 mV in 10 mV increments. (**D**) Whole-cell recording of human TMEM16F expressing cells. The macroscopic current was measured in the presence of 200 μM intracellular Ca^2+^ (upper). The representative I-V relationship of TMEM16F (lower). (**E**) Average current densities in untransfected and TMEM16C isoform-transfected 293T cells. Current density was measured at +80 mV. Data are plotted as mean + SEM (n = 3–6).

## Data Availability

The data that support the findings of this study are available on request from the corresponding author, B.-C.L.

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
