# Peer review of "Investigation of Phosphatidylserine-Transporting Activity of Human TMEM16C Isoforms"

_membranes, 2022, doi:10.3390/membranes12101005_

Round 1

Reviewer 1 Report

The manuscript from Kim et al., is an interesting work that deals with the study of the PS-transporting activity of human TMEM16C isoforms, acting as lipid scramblers which is a process involved in various physiological functions.

The techniques used are suited for the study, controls are properly done in general, and conclusions are consistent with the results obtained. There are just minor questions that authors should address before considering its publication.

-        Line 91, “Twenty-four to 48 h…”; either in letter or numbers.

-        Line 124, basal buffer units are not indicated.

-        Line 126, excitation and emission wavelengths are very close, 513 and 517 nm. Is it right? Which slits do you use to avoid excitation light entering the detector? Any filter?

-        Line 140. 146 mM of CsCl? Why do you use Cs+?

-        In figure 1, why Isoform 3 is not present in the PCR result? In the footnote or in the figure, a legend for the first column of the PCR result should be added.

-        In figure 2, the arrow indicating Ca2+ exit should be changed to Ca2+ entry.

-        In figure 2, Why Isoform 3 is not used in these experiments with GFP?

-        Line 205, it should be indicated that the study of reference 13 used a mouse tagged TMEM16C

-        Line 233 “as shown in a previous study [32]”. It should be indicated that that study was performed with fungal proteins.

-        Line 259. It is not explained how the relative surface expression ratios is calculated.

-        In figure 5, relative to the halide ion-quenching assay, I think that a positive control is needed in these experiments. For example, valinomycin is used as a positive control for similar fluorescence-based K+ assays. In the whole-cell experiments, a positive control should also be considered.

-        In the discussion section, based on their own results and/or others from bibliography, authors should discuss about the possibility that GFP could either inhibit the transport of TMEM16C protein to the surface or inhibit its function as PS-transporter. This could be clarified with a surface expression experiment as that of figure 4 but with GFP-tagged proteins.

-        Line 310. Authors say that human isoform 2 is expressed in low amount at the surface and that this result is consistent with the transport of PC and ceramide rather than PS. It is not clear for me this point. You should clarify why you relate both results.

Reviewer 2 Report

             The manuscript by H. Kim and colleagues analyzes the activity of TMEM16C isoforms as scramblases and ion channels. The protein activity was characterized by means of fluorescent microscopy, electrophysiology, Western blot methods. The results obtained under different conditions are carefully compared. The authors link the experimentally determined parameters to probable role and function of particular TMEM16C isoforms in cells.

I have several critical comments.

1) Additional details on experiments should be provided in the “Materials and Methods” section. For example, some readers definitely know that Annexin V-Alexa Fluor 568 is the fluorescent label sensible to phosphatidylserine (lines 97-98), but I’m sure that most readers are not so familiar with apoptosis observation methods. I propose to add several sentences to each paragraph in the section “Materials and methods” explaining the purpose of using a particular method or substance. It would be also useful to briefly describe the logic of each experiment (e.g., if the protein does something, then we should observe one effect; if it does not, then we should observe the other effect), and the authors’ expectations from each experimental approach.

2) Lines 125-126: “… at an excitation and emission wavelength of 513 nm and 517 nm, respectively

            The excitation and emission wavelengths are very close to each other. What was the slit width? The excitation/emission spectra of YFP should be provided.

3) Lines 172, 212, 238 and below: “DMSO (0.1%) was used as a control.”

            I’m not sure that the use of DMSO instead of ionophore A23187 provides the correct control. E.g., the ionophore itself may form pores in the plasma membrane leading to flow of PS to the outer membrane leaflet. The correct control would be an experiment with added ionophore but absent TMEM16 or with TMEM16 blocked, e.g., by specific agents. Anyway, the control should be based on experiments with ionophore present in the system.

4) Lines 259-260: “… the relative surface expression ratios of isoform 1 and 3 were 0.46 and 0.45, respectively, …

            The term “relative surface expression ratio” is not obvious. Please, explain.

5) Figure 5. The plots should be stretched along vertical axes, as it is very difficult to analyze the curves, especially, to judge on their similarity or difference. Why the current changes over time, what is the reason for such the change? Please explain/describe. In Figure 5D the vertical axis should be shortened about two times for better appearance.

Round 2

Reviewer 2 Report

The authors adequately addressed all my critical comments.